# QoS-Guaranteed Radio Resource Management in LTE-A Co-Channel Networks with Dual Connectivity

**Ren-Hung Hwang *** , **Min-Chun Peng and Kai-Chung Cheng**

Department of Computer Science & Information Engineering, National Chung Cheng University, Chiayi 600, Taiwan

* Correspondence: rhhwang@cs.ccu.edu.tw; Tel.: +886-5-272-9396



**Featured Application: Dual connectivity increases data rate of user equipment (UE), suitable for multimedia applications in cellular networks.**

**Abstract:** Dual connectivity (DC) was first proposed in 3GPP Release 12 which allows one piece of user equipment (UE) to connect to two base stations in heterogeneous networks (HetNet) at the same time, to increase the flexibility of resource utilization. DC has been further extended to multiple connectivity in 5G New Radio (NR). On the other hand, different UE tends to have different bandwidth requirements. Thus, in DC, one of the challenging issues is how to integrate resources from two base stations to enhance the quality of service (QoS) as well as the data transfer rate of each UE. In this paper, we proposed novel resource management mechanisms to improve the QoS of UE in the co-channel dual connectivity network. In terms of resource allocation, we designed the Max–Min Threshold Scheduler (MTS) which, in principle, allocates a resource block to the UE with the best channel quality while considering the issues of intercell resource allocation and the QoS requirement of each UE. In order to balance the load of different cells, we designed a novel cell selection scheme based on the HetNet Congestion Indicator (HCI) which considers not only the signal quality of UE but also the remaining resources of each base station. To improve the QoS of cell edge UE, cell range expansion (CRE) and the Almost Blank Subframe (ABS) were proposed in 3GPP. In this paper, based on Q-learning, we designed an adaptive mechanism which dynamically adjusts the ABS ratio according to the network condition to improve resource utilization. Our simulation results showed that our MTS scheduler was able to achieve a 31.44% higher data rate than the Proportional Fairness Scheduler; our HCI cell selection scheme yielded a 2.98% higher data rate than the signal-to-interference plus noise ratio (SINR) cell selection scheme; the QoS satisfaction rate of our Q-learning dynamic ABS scheme was 4.06% higher than that of the Static ABS scheme. Finally, for the cell edge users who often suffer poor data transfer rate, by integrating the mechanisms of DC, CRE, and ABS, our experimental results showed that the QoS satisfaction ratio of cell edge UEs could be improved by 10.76% as compared to the single connectivity and no ABS situation.

**Keywords:** heterogeneous network; dual connectivity; quality of service; resource allocation; cell selection

---

## 1. Introduction

Due to the rapidly growing mobile data traffic, 4G is not able to meet the current traffic demand. With the limited wireless radio resource, dual connectivity (DC) technology and heterogeneous network (HetNet) architecture can improve the data transfer rate beyond 4G. DC is based on carrier aggregation (CA) and coordinated multi-point (CoMP), allowing user equipment (UE) to connect two different base stations at the same time, which are referred to as Master eNodeB (MeNB) and Secondary

eNodeB (SeNB) [1]. Recently, DC has been further extended to include multiple connectivity in 5G New Radio (NR) [2]. The research studies on DC have attracted quite a lot of attention, e.g., in [3–5], authors have explored the potential benefits and challenges, including buffer status reporting, power headroom calculation, discontinuous reception, logical channel prioritization, bearer split, and so on. In [6], authors indicated that MeNB and SeNB have different scheduling requirements, proposing the coordination mechanism to prevent the user equipment from using beyond its maximum transmission power. In [7], a study of DC was divided into control plane and user plane issues. Using the channel state information reference signal (CSI-RS), which is unique to the Multi-input Multi-output (MIMO) frame, SeNB no longer deals with handover issues and MeNB can update the Radio Resource Control (RRC) of the user equipment to achieve handover between the different base stations. In summary, DC can effectively improve throughput in HetNet by optimal resource allocation [8]. Thus, in this paper, we focused on DC combined with the radio resource management to integrate resources across base stations and do the best to satisfy the bandwidth requirements of UEs. We explored three research topics in DC, including resource allocation, cell selection, and enhanced inter-cell interference coordination (e-ICIC).

In the study of the resource allocation, most commonly adopted resource scheduling mechanisms are Round Robin Scheduler (RR), Best Channel Quality Indicator (CQI) Scheduler (BCQI), and Proportional Fair Scheduler (PF) [9–11]. RR, which is simple to implement, assigns equal portions of resource blocks (RB) to each UE in a circular order. However, the disadvantage is that this scheduler does not consider channel condition and can lead to low system throughput. In contrast, BCQI assigns resource blocks based on the channel condition. Although the system throughput is increased, it may cause UEs with low CQI suffer starvation. PF is a compromise between throughput and fairness, based on the balance of the competing benefit among the UEs, trying to maintain fairness while maximizing throughput. Therefore, in order to emphasize the system throughput and fairness, most of the literature adopted Proportional Fair scheduling. In [12], the authors use Karush–Kuhn–Tucker optimization conditions to reduce the complexity of Proportional Fair, assuming that the transmission power of the base station is evenly distributed on each subcarrier. In [13], the geometric mean is used to replace the arithmetic mean in the Proportional Fair, which yields quicker convergence and improves throughput and block error rate (BLER). In DC, since a UE could connect to two base stations at the same time, the co-channel interference factors must also be considered. Therefore, the Proportional Fair is not necessarily the best scheduler. We proposed to estimate the channel quality of each subcarrier and assign resource block to the UE with the best CQI while considering how to integrate resources from two stations to enhance QoS as well as data transfer rate of each UE.

In the study of the cell selection, the 3GPP defines that the basic cell selection principle of the user equipment is done according to the received reference signal reception power (RSRP). The higher the transmission power of the base station, the wider the range of signals that can be covered and the more UEs will connect to that base station [14,15]. Many studies indicated that the RSRP is conducive to large base stations (e.g., Macro BS), but not obvious for small cells (e.g., pico BS). In [16], the cell range expansion (CRE) scheme is investigated, in which the coverage of pico BS is extended by adding additional reference signal such that more UEs could select the pico BS rather than Macro BS. However, increasing the CRE bias value alone does not improve the performance of the small cell, as the cell edge users are significantly affected by the intercell interference. In [17], the serving cell is selected according to the signal-to-interference plus noise ratio (SINR) which considers not only the transmission power of the base station, but also the interference between the base stations. Many other factors are also considered in cell selection in the literature, such as load balance, utility function of proportional fair, dynamic strategy, and data transfer rate [18–21]. In this paper, based on the special features of DC, we consider load balancing and downlink data transfer rate in the cell selection. Specifically, we first estimate CQI based on SINR and then estimate the number of resource blocks that a UE can be assigned, and finally calculate the actual data transfer rate.

In the study of the enhanced intercell interference coordination, some previous works took the Almost Blank Subframe (ABS) into consideration [22,23], in which a ratio of time slots of a frame of the Macro BS were preserved for the pico BS to achieve time domain coordination. Most of previous works adopted static ABS setting and showed that the ratio of 0.5 yielded the best performance. However, it may waste a lot of wireless resources of the Macro BS. In this paper, we proposed a better method to adjust the ABS ratio dynamically according to the network conditions. It would decrease the ABS ratio when the Macro BS is overloaded, and increase the ABS ratio to improve the throughput of pico BS when the load of Macro BS is low. Simulation results are given to substantiate theoretical findings with a comparison with fixed ABS configuration.

Besides, there are various types of small cell and UEs with different bandwidth requirements in HetNet. Most literature focused on single traffic, which is not able to reflect the real-world situation [24]. Thus, in this work, multiple types of applications with different QoS requirements were considered. By considering voice traffic and video traffic, we classify two types of the traffic into different QoS indicators: Guaranteed bit rate (GBR) and maximum bit rate (MBR). The QoS requirement of GBR is a guaranteed fix transmission rate, while that of MBR is a guaranteed of minimum rate and a limit of maximum rate. The numerical results show that our proposed schemes had significant improvement in the QoS satisfaction ratio than previous works in the literature.

The rest of this paper is organized as follows. Section 2 introduced system design. The proposed Max–min Threshold Scheduler (MTS), HetNet Congestion Indicator (HCI), and Q-learning dynamic ABS will be described in detail in Section 3. The system deployment and the simulation results present in Section 4. Finally, in Section 5, conclusion and future works are discussed.

## 2. System Model

### 2.1. Co-Channel Networks with Dual Connectivity

For HetNets, the wireless radio resources not only determine the quality of the network but also determine whether the bandwidth requirements of UEs are satisfied. In the limited wireless radio resources, how to effectively improve the spectrum utilization is very important; therefore, co-channel HetNet is currently the most efficient approach, meaning that Macro BS and pico BS co-use the same frequency. However, the challenge of the co-channel network is the interference between each base station. Based on the different downlink transmission power of the base station in HetNets, the low power pico BS with the distance closer to the Macro BS will suffer the greater interference, and lead to the decrease of the signal coverage of the pico BS, as shown in Figure 1 [25].

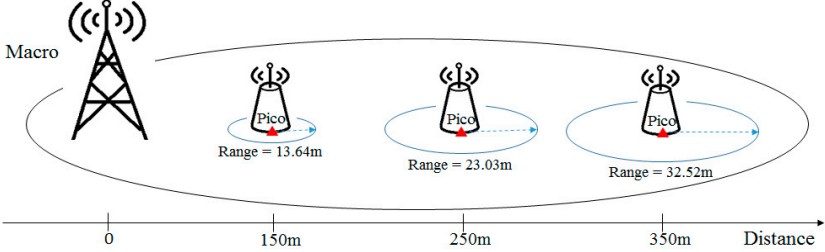

**Figure 1.** Pico base station (BS) coverage area in the presence of a Macro BS.

In this paper, we combine the dual connectivity technology with CRE and ABS by integrating resources from two base stations to enhance the data transfer rate of each UE. For example, in 3GPP co-channel network scenario [1] as shown in Figure 2, it can be observed that UE 1 and UE 3 are close to the pico BS and Macro BS, respectively, and can receive better CQI than the UE 2. On the other hand, UE 2 is located at the signal junction of the base station and has a worse CQI due to the greater interference. In order to enhance the data transfer rate, UE 2 with dual connectivity can access the resources from Macro BS in non-ABS situations and can also access the resources from pico BS in ABS situations (see part (a) in Figure 2) where non-ABS means the ABS mechanism is not enabled.

We learned that the difference between the dual connectivity and the original single connectivity (SC) is that some UEs, such as UE 2, can access radio resources from the Macro BS and the pico BS simultaneously. In order to facilitate the management, as shown in Figure 2b, we define two categories of the user equipment, namely Macro user equipment (MUE) and Pico user equipment (PUE). The characteristics of each category are described as follows.

- **Macro user equipment (MUE):** Only access the radio resources from the Macro BS, and cannot access the radio resources from the pico BS. Therefore, in our proposed mechanisms, we will give the MUE a higher priority than the PUE in the resource allocation of the Macro BS.
- **Pico user equipment (PUE):** Besides accessing the radio resources from the pico BS, a UE may also access the radio resources from the Macro BS.

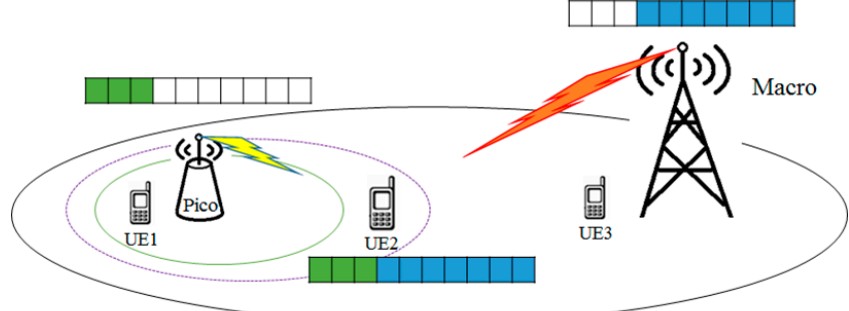

(a) dual connectivity combined with CRE and ABS

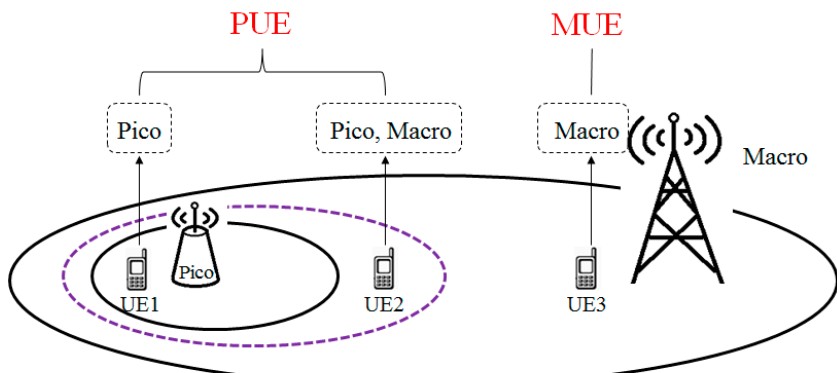

(b) define the categories of the user equipment

**Figure 2.** Dual connectivity conditions.

*2.2. Optimal Resource Allocation*

In LTE-A, the basic schedule unit is a resource block (RB). One RB consists of 12 subcarriers and a time domain, and the base station may calculate the data transfer rate of an RB when it is allocated to a specific UE. Due to the difference in channel quality, the signal status of different UE is different, so the data transfer rate will be affected by the modulation and coding rate (see Table 1).

**Table 1.** Channel Quality Indicator (CQI) table.

| CQI Index | Modulation | Code Rate × 1024 | Efficiency |
|:---:|:---:|:---:|:---:|
| 0 | | out of range | |
| 1 | QPSK | 78 | 0.1523 |
| 2 | QPSK | 120 | 0.2344 |
| 3 | QPSK | 193 | 0.3770 |
| 4 | QPSK | 308 | 0.6016 |

**Table 1.** *Cont.*

| CQI Index | Modulation | Code Rate × 1024 | Efficiency |
|:---:|:---:|:---:|:---:|
| 5 | QPSK | 449 | 0.8770 |
| 6 | QPSK | 602 | 1.1758 |
| 7 | 16QAM | 378 | 1.4766 |
| 8 | 16QAM | 490 | 1.9141 |
| 9 | 16QAM | 616 | 2.4063 |
| 10 | 64QAM | 466 | 2.7305 |
| 11 | 64QAM | 567 | 3.3223 |
| 12 | 64QAM | 666 | 3.9023 |
| 13 | 64QAM | 772 | 4.5234 |
| 14 | 64QAM | 873 | 5.1152 |
| 15 | 64QAM | 948 | 5.5547 |

The impact of the resource scheduling on the data transfer rate is very significant. Since all UEs share bandwidth of a base station, to optimize the system throughput it is intuitive to assign an RB to the UE with the best channel quality. That is, we formulate a throughput optimization problem as follows.

$$\max_{X_{u,k}} \sum_{u \in U} \sum_{k=1}^{K} TP_{u,k} * X_{u,k} \tag{1}$$

where $u$ is a UE belongs to the UE set $U = \{u_1, \ldots, u_U\}$, $k$ is a RB of the RB set $K = \{1, 2, \ldots, K\}$, $TP_{u,k}$ is the throughput of RB $k$ when it is allocated to the UE $u$, and $X_{u,k}$ represents whether RB $k$ is assigned to UE $u$. $TP_{u,k}$ can be calculated by 12 (subcarriers) * 7 (symbols) * $CQI_{efficiency}$. There is a positive correlation between $TP_{u,k}$ and $CQI_{efficiency}$; however, $CQI_{efficiency}$ is not a linear function of $CQI_{index}$, which makes the objective function nonlinear. Therefore, we replace $CQI_{efficiency}$ with $CQI_{index}$ and make the optimization problem (1) to a 0–1 integer programming problem as follows:

$$\max_{X_{u,k}} \sum_{u \in U} \sum_{k=1}^{K} CQI_{u,k} * X_{u,k}. \tag{2}$$

Subject to:

$$\sum_{u \in U} X_{u,k} \leq 1 \forall k \tag{3}$$

$$X_{u,k} \in \{0, 1\} \forall u, k \tag{4}$$

where Equation (3) constrains each RB can only be assigned to a UE, and Equation (4) states that $X_{u,k}$ is an indication variable.

Two common approaches of solving 0–1 linear programming problems are "implicit enumeration" and "branch and bound". Unfortunately, the computational complexity of these two methods is exponential time. In order to reduce the computational complexity, we can first relax the 0–1 integer programming to linear programming. The relaxed linear programming can then be solved by many existing solutions, such as "simplex method", "ellipsoid method", and "interior point method". For example, in MATLAB, the linprog library is provided which can obtain the optimal solution by the "internal point method" according to the scale of the optimization problem.

After obtaining the optimal solution from linear programming, two methods can be used to transfer the solution back to the 0–1 integer programming solution.

- **Max method:** For RB $k$, the sum of all variables is at most 1 under constraint (3). The max method sorts these variables in descending order, assigns 1 to the variable with the largest value (i.e., allocate the RB to the UE) and 0 to the rest variables.
- **Random method:** For RB $k$, let $N = \sum X_{i,k}$. The random method allocates RB $k$ to UE $i$ with probability $p_i$ where $p_i = \frac{X_{i,k}}{N}$ (i.e., set $X_{i,k} = 1$ and $X_{u,k} = 0 \ \forall u \neq i$).

As aforementioned, we will propose a max–min RB allocation method which will be described in detail in the next section. In the following, we performed experiments to compare the performance of three RB allocation methods, namely, the max method, the random method, and the max–min method. We compared three performance metrics: The similarity between the optimal solution (of the 0–1 integer programming) and that of these methods, the difference of objective function (1) and (2) of the optimal solution, and that of these methods. The similarity was defined as the pair-wise comparison of each variable of the optimal solution, and that of the three methods. For example, if the system had 50 RBs, and allocation of 40 RBs were the same between two methods, we would define the similarity as 80%. For the objective function, we compared the degradation of the objective function of one method as compared to the optimal solution. For example, 10% means the objective function obtained by one approximate method only achieves 90% of that of the optimal solution.

The system parameters of the experiments are shown in Table 2, in which the system has seven Macro BSs, each Macro BS contains three sectors, and each sector deploys five user equipments. We simulate 500 TTIs, and for each TTI, the optimal solution as well as the solutions of the three methods were calculated for each sector (i.e., 21 sectors).

**Table 2.** Simulation parameter of the optimal resource allocation.

| Parameters | Configuration |
| --- | --- |
| Simulation time | 500 TTI |
| Cellular layout | Wrap around, 7 Macros, 3 sectors per site |
| Carrier frequency | 2.14 GHz |
| Bandwidth | 10 MHz |
| Number of RBs | 50 |
| Number of UEs per sector | 5 |
| Macro transmit power | 46 dBm |
| Thermal noise | −174 dBm/Hz |
| Pathloss model | As in 3GPP TS 36.942 |

Table 3 shows the experimental results. As can be observed, random method did not perform well, neither yielded good similarity to the optimal solution nor achieved good objective value. The max method yielded the closest solution to the optimal solution. On the other hand, the max–min method also yielded a very competitive solution to the max method, in particular, both methods yielded very high objective value with less than 1% degradation. However, the max-min method had much less computation complexity as compared to the other methods which need to solve 0–1 integer linear programming or linear programming problem. In summary, the max-min method is a good candidate mechanism for RB allocation.

**Table 3.** Experimental results.

| | Similarity | Degradation of Obj. Func. (1) | Degradation of Obj. Func. (2) |
| --- | --- | --- | --- |
| Max method | 93.96% | 0.43% | 0.67% |
| Random method | 36.03% | 37.99% | 48.42% |
| Max–Min method | 87.62% | 0.47% | 0.83% |

*2.3. Guarantee Quality of Service (QoS)*

In order to evaluate the system performance such as throughput, most literature assume UEs have full buffer. However, in reality, each UE has its own traffic characteristic and quality of service requirement. Allocating resources more than a UE's needs does not increase system throughput. In this work, we considered two types of traffic, each with its own bandwidth requirement. By considering voice traffic and video traffic, we classified two types of the traffic into two QoS categories: Guaranteed bit rate (GBR) and maximum bit rate (MBR). The characteristics of each QoS requirement are described as follows [26]:

- **Guaranteed bit rate (GBR):** Some applications, such as audio, require a fixed amount of bandwidth continuously during their lifetime. Thus, the traffic characteristic as well as QoS requirement of this kind of traffic is a constant data rate.
- **Maximum bit rate (MBR):** Yet, other types of applications, such as video, require a variable amount of bandwidth. The traffic may be bursty, and is usually characterized by a peak data rate and a sustainable data rate. Thus, we set the QoS requirement of MBR with two parameters, namely a minimum of the sustainable data rate (or average data rate) and a peak (maximum) data rate.

We took the QoS requirement into consideration while scheduling RBs. That is, when a base station allocates RBs to UEs under its coverage, it will try its best to guarantee the QoS of each UE under the constraint of limited RBs. Specifically, to provide QoS guarantee to UE *u*, the number of RBs to be allocated depends on its QoS requirement and channel condition. For GBR traffic, the required data rate is its guaranteed data rate while for MBR traffic, it is its minimum data rate. The number of required RBs for UE *u* can be calculated by following equation [24]:

$$NRB_u = \left\lceil \frac{UE_{u,QoS}}{RB_{data\_rate}} \right\rceil \tag{5}$$

where $UE_{u,QoS}$ is QoS requirement for UE *u*, $RB_{data\_rate}$ is data rate of a RB, and $NRB_u$ is the minimum number of the required RBs to satisfy the QoS of UE *u*. If the base station has abundant RBs, it will not allocate more RBs to UEs with GBR traffic, instead, it will allocate to UEs with MBR traffic, but only up to their peak data rate.

## 3. Proposed Schemes

The radio resource management scheme of base stations affects the performance of applying dual connectivity in HetNets. In this section, we propose three novel schemes in resource scheduling, cell selection, and enhanced intercell interference coordination, which are named Max—min Threshold Scheduler (MTS), HetNet Congestion Indicator (HCI), and Q-learning Dynamic Almost Blank Subframe (QD-ABS), respectively. The details for these schemes are outlined in the following.

### 3.1. Max–Min Threshold Scheduler

The MTS approach can be split into two different views: max–min scheduling and Threshold restrictions. The former is responsible for optimizing the resource allocation, the latter limits the allocation of RBs according to the environmental conditions with dual connectivity.

3.1.1. Max–Min Scheduling

For one RB, the achievable data transfer rate can be calculated by base station when the RB is assigned to a UE. In general, the base station will allocate an RB to the UE with the best channel quality while considering the issues of intercell resource allocation and the UE's QoS requirement. We assumed that there are N RBs within a time slot and *M* UEs connected to the base station. Based on UE's CQI, the base station constructs an NxM matrix *A* as shown in Figure 3. The max–min scheduling is then used to assign RBs to UEs to satisfy QoS of UEs as much as possible.

The max–min scheduling consists of five steps:

(1) Find the maximum (max) of the CQI value in the matrix *A*, and assume it is *A(i, j)*. It assigns the *i*-th RB to the *j*-th UE so that the UE can receive the highest data rate with this RB.

(2) When there are more than one maximum CQIs in the matrix *A*, it selects the RB with minimum (min) impact to the other UEs in the same row. For example, if *A(i1, j1)* and *A(i2, j2)* are the same maximum in the matrix *A*, then it calculates $s(i1) = \sum_{j=1}^{M} A(i1, j)$ and $s(i2) = \sum_{j=1}^{M} A(i2, j)$, respectively. It then assigns the *i*1-th RB to the *j*1-th UE first if $s(i1) < s(i2)$.

(3)    If $s(i1) = s(i2)$, it sorts the vector $A(i1,*)$ and $A(i2,*)$ in descending order, compares the elements one by one until it finds the one with less value, say it is $A^s(i1,j)$. It then assigns the $i1$-th RB to the $j$-th UE (where $A^s(i1,*)$ is the sorted vector).

(4)    If all the elements are still the same, it randomly selects one, say $A(i1, j1)$, and assigns the RB to a UE accordingly.

(5)    After allocating the $i$-th RB, the $i$-th row of the matrix $A$ is cleared to 0 (indicating that the RB had been allocated).

The max-min scheduling repeats the step from (1) to (5) until all of the UEs have satisfied QoS requirement or all of the RBs had been allocated.

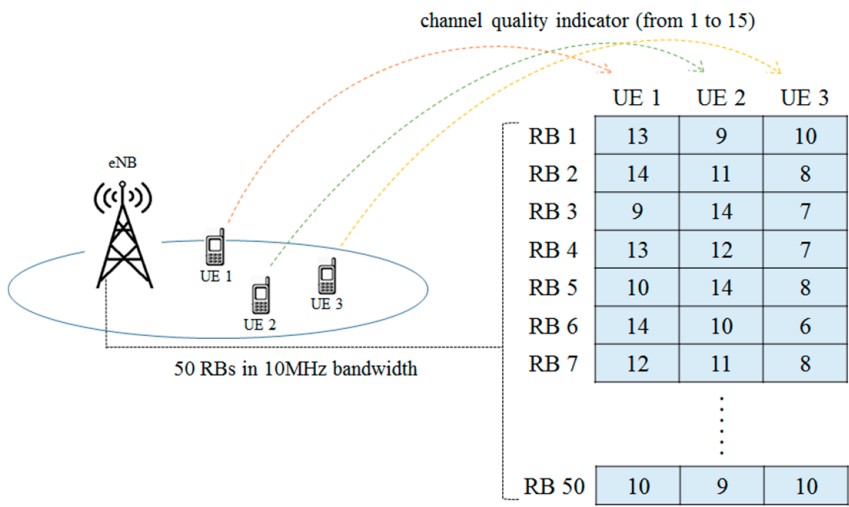

**Figure 3.** Table of user equipment (UE)'s CQI at each resource block (RB).

Once the RB allocation is completed, the max–min scheduling then evaluates whether the QoS requirements of the currently assigned UEs are satisfied. In LTE-A, the transport block (TB) is used to calculate the throughput of the downlink. A TB consists of several RBs, and all RBs must be transmitted using unified CQI encoding. Given the number of available RBs that can be assigned to the UE, it calculates the maximum data rate it can receive from the base station using following equation:

$$TB = N_{symbol} * RB_{assigned} * CQI_{efficiency} * T_{slot} - CRC \tag{6}$$

where $N_{symbols}$ is the number of symbols per RB, $RB_{assigned}$ is the number of RBs which the UE can be assigned, $CQI_{efficiency}$ is the bits per RB, $T_{slots}$ is the number of two time slots per millisecond, and $CRC$ is the cyclic redundancy check in 24 bits.

### 3.1.2. Threshold Restrictions

Recall that there are two types of UEs in the system, namely MUE and PUE. Thus, we let $U = U^M \cup U^P$, where $U^M$ and $U^P$ are the set of MUEs and PUEs respectively. Since MUE can only access RBs of the Macro BS, the proposed MTS scheme gives higher priority to MUE when allocating RBs of the Macro BS, as illustrated in Figure 4. The MTS adopts a threshold-based scheme which allocates a certain number of RBs to MUEs before it exercises the max–min scheduling. The basic idea is to guarantee the QoS of MUEs first. It then adopts the max–min scheduling to allocate the remaining RBs to both MUEs and PUEs. On the other hand, the pico BS exercises the max-min scheduling to allocate the RBs to PUEs only.

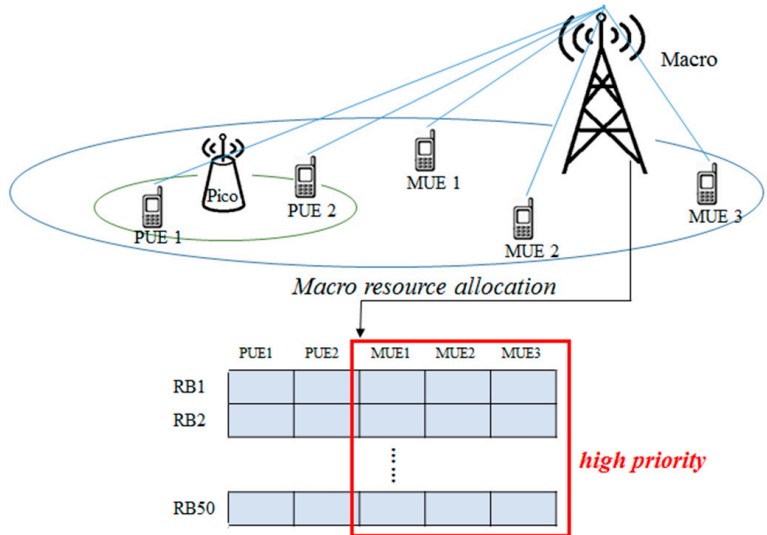

**Figure 4.** Macro BS adopts threshold-based RB allocation before applying the max-min scheduling.

The number of RBs reserved for MUEs is based on solving following optimization problem, similar to the problem defined in Section 2, but two new constraints are added, as shown in the following.

$$\max_{X_{u,k}} \sum_{u \in U^M} \sum_{k=1}^{K} CQI_{u,k} * X_{u,k}. \tag{7}$$

Subject to:

$$\sum_{u \in U^M} X_{u,k} \leq 1 \quad \forall k \tag{8}$$

$$\sum_{k=1}^{K} X_{u,k} \leq NRB_u \quad \forall u \in U^M \tag{9}$$

$$CQI_{u,k} \geq 7 \quad \forall k \tag{10}$$

$$X_{u,k} \in \{0, 1\} \quad \forall u \in U^M, k \tag{11}$$

where Equation (9) means that the number of allocated RBs cannot exceed the QoS requirement of the MUE, and Equation (10) means that the channel quality must be higher than CQI 7 to avoid wasting RBs on low CQI MUEs. In summary, the threshold-based mechanism is expected to achieve similar performance of the max-min scheduling because both methods are based on similar optimization objectives. The purpose of the threshold-based mechanism is to protect MUEs to access enough resources before competing with PUEs.

*3.2. HetNet Congestion Indicator*

Due to the dual connectivity, each piece of user equipment is connected to the Macro BS. The handover problem between the HeNB will be processed by the Macro BS. We propose a new HetNet Congestion Indicator (HCI) cell selection scheme to achieve the load balance, considering not only the signal quality of the UE, but also the number of remaining resources of each base station. In addition to the cell selection, we also use HCI to adjust the type of the UE, either MUE or PUE.

3.2.1. Cell Selection Scheme

When a UE has more than one BS to join, it selects the one with least HCI. The key idea of HCI is to estimate the congestion level of a base station based on the competitive online algorithm proposed in [27]. Specifically, the HCI cell selection scheme consists of two steps:

(1) A UE continuously measures the SINR among neighboring cells. If the SINR of current serving BS becomes weaker and some neighboring BSs become stronger, then the UE enters the cell selection procedure (step 2).

(2) The UE selects a list of candidate BSs which have higher SINRs. For each BS, the UE estimates and reports its Channel Quality Indicator (CQI) to the BS. By taking account of the QoS requirement of the UE and the status of RB usage, the BS replies to the UE with its HCI which is calculated as follows. Assume the maximum number of RBs per time slot (system capacity) is $C$, number of RBs which have been used (allocated) is $R_i$, the number of RBs required by the UE (based on its CQI and QoS requirement) is $b$, the HCI is given by:

$$HCI(b) = cost(R_i + b,\ C) - cost(R_i, C) \tag{12}$$

where the *cost()* function is defined as follows.

$$cost(R_i, C) = \mu^{(\frac{R_i}{C} - 1)} \tag{13}$$

where $\mu$ is a constant parameter which is usually set to $C$ *(in our case, it is set to 50)*.

After obtaining the HCI of each candidate BS, it selects the BS with the least HCI to join. The HCI function is meant to estimate the congestion level or traffic load of a BS.

### 3.2.2. Classify a UE as a MUE or PUE Type

As mentioned in Section 2, a UE could be either MUE or PUE. If a UE receives signal from both Macro BS and pico BS, and it selects the Macro BS as its serving BS as the BS has the least HCI, then the UE is classified as a MUE; otherwise, it is a PUE. Due to the mobility, a UE may change its serving BS which will also change its type. As aforementioned, the different type of the UE will affect the priority of accessing resources. Specifically, a MUE can only access resources from the Macro BS, while a PUE can access resources from two BSs simultaneously. Figure 5 illustrates how a UE may enter the dual connectivity range or transits from PUE to MUE. Initially, a UE may be within the coverage range of a pico BS, but far from the Macro BS. In this stage, it is a PUE. As it moves closer to the Macro BS, it begins to receive signal from Macro BS and enters the dual connectivity range. At this stage, it is able to utilize resources of both BSs. As it continues to move closer the Macro BS, it eventually selects Macro BS as its serving BS based on the HCI cell selection scheme. At this stage, it becomes a MUE.

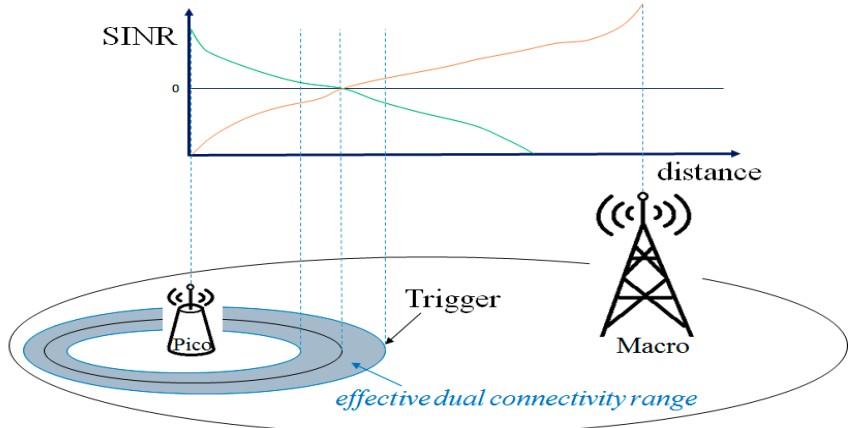

**Figure 5.** Signal coverage in dual connectivity mode.

### 3.3. Q-Learning Dynamic Almost Blank Subframe

In order to explore the dual connectivity mechanism combined with CRE and ABS, the ABS configuration learning procedure is introduced in this section. We propose a novel Q-learning dynamic

ABS method based on reinforcement learning. Furthermore, it is effective to adjust the ABS ratio and reduce the interference in the co-channel networks by coordinating the adjacent Macro BSs. Thus, we also propose a centralized decision to adjust the ABS ratio which requires all Macro BSs to synchronize their ABS ratio.

### 3.3.1. Related Works

Prior works on ABS ratio can be classified into fixed and dynamic ABS ratio settings. In the fixed dynamic ABS ratio setting schemes, ABS ratio is set to a fixed value based on some experiments, e.g., [28,29]. On the other hand, the dynamic ABS ratio setting schemes adjust the ABS ratio according to the system status, such as traffic load and number of edge UEs. Quite a few of dynamic ABS ratio setting schemes are based on reinforcement learning [30–32].

Reinforcement learning is a machine learning method which focuses on issues such as learning with feedback or sequential decision-making. The purpose is to let the machine learn actively what kind of action can be achieved with the maximal feedback, rather than tell the machine what action to take [30,31]. Different from the supervised learning, reinforcement learning can occur without a teacher by repeatedly interacting with the environment. It does not require accurate input and output, emphasizing a continuous decision-making action of the online planning. For example, Figure 6 illustrates the structure of ABS learner–environment interaction. It is difficult to adjust the ABS ratio in the unknown environment, which consists of factors such as the designed cellular layout, the density of the connected UE, interference between the BSs, and so on. However, in such a complex situation, reinforcement learning will allow the system to select an arbitrary ABS ratio first, observe the feedback of whether the performance is good or not, and then adjust (learn) the ABS ratio according to the feedback. The system continues to observe and adjust until the most suitable ABS ratio is learned.

Q-learning is a well-known algorithm for reinforcement learning. The Q-learning consists of the various states, action, and cost. The agent of Q-learning will continue to learn until a task is reached. In other words, the agent learns how to choose the action at different states to achieve the best reward or the minimum punishment by repeatedly interacting with the environment.

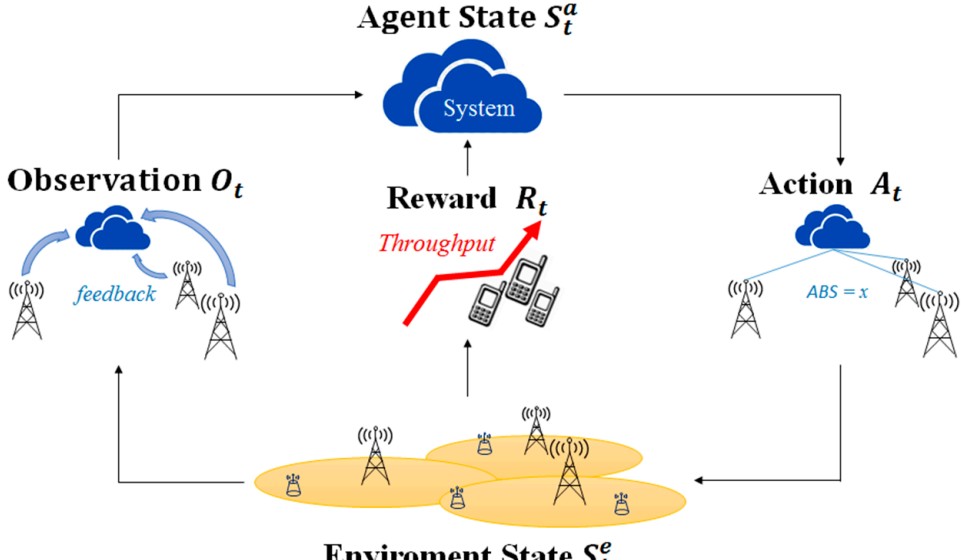

**Figure 6.** The Almost Blank Subframe (ABS) learner-environment interaction.

### 3.3.2. Q-Learning-Based Dynamic ABS (QD-ABS) Scheme

In this work, we proposed a Q-learning-based dynamic ABS scheme to learn the appropriate ABS configuration for a global ABS ratio setting. The key idea is inspired from the literature [33]. In [33], the ABS ratio was dynamically adjusted based on the throughput of Macro BS and pico BS. Since the UE distribution in each sector of a macro station is different, the ABS ratio will also be different. Furthermore, the difference in ABS ratios will affect the throughput of each BS. Thus, we can define the difference between the distribution of the throughput as the cost parameter in Q-learning. The agent finally chooses the best ABS ratio when Q-learning is converged to the minimum cost.

The detailed steps of Q-learning-based dynamic ABS scheme consists of two steps:

(1) It estimates the required ABS ratio in each sector of the Macro station first. Let the total bandwidth requirement of the UEs in Macro BS and pico BS be $N_{MUE}$ and $N_{PUE}$, respectively. Assume that the sum of the throughput allocated by the Macro BS and pico BS in the previous frame to the UEs is $Th_{MUE}^*$ and $Th_{PUE}^*$, respectively. Thus, the difference between the required bandwidth and the actual assigned throughput can be expressed as follows.

$$\Delta Th_{MUE} = Th_{MUE}^* - N_{MUE} \tag{14}$$

$$\Delta Th_{PUE} = Th_{PUE}^* - N_{PUE}. \tag{15}$$

In order to meet the QoS requirements of the UE, the Equation (16) illustrates the opportunity of using ABS, which contains two conditions. The first one is that all of the MUEs must satisfy QoS requirement, and there are remaining resources in the Macro BS ($\Delta Th_{MUE} > 0$). The other one is that some PUEs have not satisfied QoS requirement after resource allocation in pico BS ($\Delta Th_{PUE} < 0$). Let the set of Macro BSs be denoted by $M$. If the above two conditions are met, then we can first give the parameter $\mu_b$ ($b \in M$) as the ABS ratio in each sector of the Macro station.

$$ABS \ ratio \begin{cases} \mu_b, if \Delta Th_{MUE} > 0 \text{ and } \Delta Th_{PUE} < 0 \\ 0, \qquad others. \end{cases} \tag{16}$$

Assuming that the throughput of the radio resource is evenly distributed across each TTI in a frame. For example, the allocation of 1000 bits by a frame is equivalent to an average transmission of 100 bits in 10 TTI. The formula of the ABS $\mu_b$ is described as follows:

$$\mu_b = \underset{\mu_b}{\operatorname{argmax}} \begin{cases} (1 - \mu_b) Th_{MUE}^* - N_{MUE} \geq 0 \\ [1 + \mu_b (\varepsilon - 1)] Th_{PUE}^* - N_{PUE} \leq 0 \\ 0 < \mu_b < 1 \end{cases} \tag{17}$$

where $\varepsilon$ is the gain factor and $\mu_b (\varepsilon - 1)$ is the ratio of increased data transfer rate of the PUE after using ABS over the original data rate. Through the difference in the throughput and the restriction of the QoS requirement, ABS $\mu_b$ can be calculated by solving the simultaneous inequalities. Equation (17) consists of three conditions: (1) The decreased throughput of MUEs will not be less than their required bandwidth ($N_{MUE}$); (2) the increased throughput of PUEs will not be more than their required bandwidth ($N_{PUE}$); (3) the value of $\mu_b$ must be between 0 and 1. The aim is to avoid wasting the resources of the MUE after using ABS.

(2) With the ABS $\mu_b$ of each sector of the Macro station, the next step is to learn the unified ABS ratio by Q-learning algorithm. The state, action, and cost of the Q-learning algorithm based on the ABS configuration are defined as follows:

- **Agent**: The system.
- **State**: The state is defined as $s$, which is the satisfaction ratio of PUEs in the system. The satisfaction ratio of PUEs is defined as the number of PUEs whose QoS requirements can be met over the total number of PUEs.

- **Action**: The action is defined as $a_t$, which is the ABS ratio.
- **Cost**: The cost is defined as $c$, which is the new calculated throughput of each sector of the Macro BS after using the unified ABS ratio.

The cost $c$ estimates the immediate return incurred due to action $a_t$ at state $s$. The cost function is calculated as:

$$\Delta\mu = a_t - \mu_b \tag{18}$$

$$Cost_b \begin{cases} 0, & if\,\Delta\mu = 0 \\ \left|\Delta\mu Th_{MUE}{}^*\right|, & if\,\Delta\mu > 0 \\ \left|\Delta\mu\,(\varepsilon - 1)\,Th_{PUE}{}^*\right|, & if\,\Delta\mu < 0 \end{cases} \tag{19}$$

$$c = \sum Cost_b \tag{20}$$

where Equation (18) represents the difference before and after adjusting the ABS ratio. In Equation (19), $\Delta\mu > 0$ indicates that the actual ABS ratio chosen by the system ($a_t$) is greater than that of the sector of the Macro BS estimated by itself ($\mu_b$), thus losing the excessive throughput of MUEs. On the contrary, $\Delta\mu < 0$ indicates that the actual ABS ratio chosen by the system is less than that of the sector of the Macro BS estimated by itself, thus losing the expected throughput of PUEs. Finally, the sum of these costs in each sector of the Macro BS is calculated in Equation (20).

Given aforementioned parameters, the Q-value $Q(s_t,\ a_t)$ can be defined as the expected cost of taking action $a_t$ at state $s_t$. Then Q-value is updated as follows:

$$Q(s_t,\ a_t) = (1-\rho)Q(s_t,\ a_t) + \rho\left[c + \min_{a_{t+1}}\{Q(s_{t+1},\ a_{t+1})\}\right] \tag{21}$$

where $s_{t+1}$ is the next state after taking the action $a_t$, and $\rho$ is the learning rate, which denotes the willing to learn from the environment. Eventually, a Q-table will be built as shown in Table 4. Once these values in a Q-table have been learned and converged, the optimal action at each state is the one with the lowest Q-value.

**Table 4.** Q-table in Q-Learning-Based Dynamic ABS (QD-ABS) scheme.

| | | **Dynamic ABS Ratio** | | | | |
|---|---|---|---|---|---|---|
| | | $a_1$ | $a_2$ | $a_3$ | . . . | $a_n$ |
| **PUE satisfaction ratio** | < 25% | $Q(s_1,a_1)$ | $Q(s_1,a_2)$ | $Q(s_1,a_3)$ | . . . | $Q(s_1,a_n)$ |
| | 25~50% | $Q(s_2,a_1)$ | $Q(s_2,a_2)$ | $Q(s_2,a_3)$ | . . . | $Q(s_2,a_n)$ |
| | 50~75% | $Q(s_3,a_1)$ | $Q(s_3,a_2)$ | $Q(s_3,a_3)$ | . . . | $Q(s_3,a_n)$ |
| | > 75% | $Q(s_4,a_1)$ | $Q(s_4,a_2)$ | $Q(s_4,a_3)$ | . . . | $Q(s_4,a_n)$ |

The detailed pseudo-code is given as follows:

---

**Algorithm: Q-learning dynamic ABS (QD-ABS)**

---

    **Initialize:**

1   **for** each $b \in M$ **do**

2      calculate the initial ABS ratio $\mu_b$ based on Equations (23) and (24)

3   **end for**

4   **for** each $s \in S$, $\alpha \in A$ **do**

5      initialize the Q-value as the $Q(s_t, \alpha_t)$

6   **end for**

    **Learning:**

7   **loop**

8      get the current state $s_t \in S$

9      select the action $a_t \in A$ which has the minimum Q-value

10   calculate $\Delta \mu$ based on Equation (18)

11   **for** each $b \in M$ **do**

12     calculate the cost of changing the ABS ratio from $\mu_b$ to $\alpha_t$ based on Equation (19)

13     sum the cost to $c$ based on Equation (20)

14   **end for**

15   observe the next state $s_{t+1}$ and update the Q-table entry as follows:

16   $Q(s_t, a_t) = (1 - \rho)Q(s_t, a_t) + \rho \left[ c + \min_{a_{t+1}}\{Q(s_{t+1}, a_{t+1})\} \right]$

17   **end loop**

18   select the action $\alpha_t$ which has the minimum Q-value for state $s$

---

QD-ABS algorithm

## 4. Evaluation

### 4.1. Simulation Assumption

In this work, we adopted the system level simulation [34] to validate our proposed schemes. Specifically, we used Vienna LTE-A Downlink System Level Simulator version v2.0 Q3-2018 as our simulator [35]. The LTE-A downlink macro-pico HetNet with seven cell sites wrapped around and three hexagonal sectors per cell site was considered here. In each cell, two pico BSs with fixed position were deployed at the cell edge with a distance of 8/15 ISD (Inter-site Distance) from the macro BS, as shown in Figure 7. In addition, mobile UEs that were nonuniform were distributed in the network. The total number of UEs was 315 in the simulated scenario with 15 UEs deployed in each sector of a Macro BS. Two traffic types were considered among 315 UEs, 158 UEs generating GBR traffic and 157 UEs generating MBR traffic. Each simulation was run for 10,000 TTIs and 30 runs of simulations were conducted. The 95% confidence interval of each simulation case was also calculated. System parameters of the simulation are summarized in Table 5.

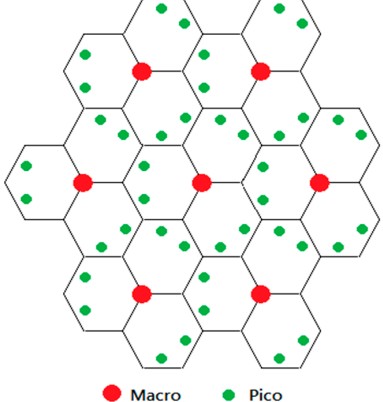

**Figure 7.** Layout of network topology.

**Table 5.** System parameters of the simulation.

| Parameter | Configuration |
|---|---|
| Cellular layout | Wrap around, 7 Macros, 3 sectors per site |
| Carrier frequency | 2.14 GHz |
| Bandwidth | 10 MHz |
| ISD | 500 m |
| Macro transmit power | 46 dBm |
| Pico transmit power | 30 dBm |
| Number of picos per sector | 2 |
| Number of UEs per sector | 15 |
| Cell selection | HCI and SINR |
| Scheduler | MTS and PF |
| ABS | Q-learning dynamic and Static |
| Thermal noise | −174 dBm/Hz |
| UE speed | 3 km/hr |
| Pathloss model | As in 3GPP TS 36.942 |
| Traffic model | VoIP (GBR for 1.5 Mbps) and Video (MBR for 3 Mbps–7 Mbps) |
| Duplex model | FDD |
| Number of RBs | $C = 50$ |
| Learning rate | $\rho = 0.5$ |

### 4.2. Q-Learning-Based Dynamic Almost Blank Subframe

For performance comparison of our proposed schemes, the following simulation scenarios were considered (see Table 6). The purpose of designing the five scenarios is illustrated as follows.

- **Scenario 1**: It is designed to verify the performance of the HCI cell selection scheme, which is compared with the SINR scheme.
- **Scenario 2**: It is designed to verify the performance of the MTS resource allocation scheme, which is compared with the Max-min scheme.
- **Scenario 3:** It is designed to verify the performance of the PF resource allocation scheme in dual connectivity environment, which is compared with our proposed MTS scheme. In addition, it is also compared with itself in single connectivity environment.
- **Scenario 4:** It is designed to verify the performance of the QD-ABS scheme in single connectivity, which is compared with the static ABS scheme.
- **Scenario 5:** It is designed to verify the performance of dual connectivity combined with CRE and ABS.

**Table 6.** Simulation scenarios.

| | Connection | Cell Selection | Scheduler | E-ICIC | CRE Bias |
|---|---|---|---|---|---|
| Scenario 1 | DC | HCI<br>SINR | MTS | Non-ABS | 0 dB |
| Scenario 2 | DC | HCI | MTS<br>Max-min | Non-ABS | 0 dB |
| Scenario 3 | DC | HCI | MTS<br>PF | Non-ABS | 0 dB |
| | SC | HCI | PF | Non-ABS | 0 dB |
| Scenario 4 | SC | SINR | PF | Non-ABS<br>QD-ABS<br>Static ABS = 0.5 | 0 dB<br>6 dB<br>6 dB |
| Scenario 5 | SC | SINR | PF | Non-ABS | 0 dB |
| | SC | SINR | PF | Static ABS = 0.5 | 6 dB |
| | DC | HCI | MTS | Non-ABS | 0 dB |
| | DC | HCI | MTS | QD-ABS | 6 dB |

### 4.3. Output Indicators

In this work, the QoS requirement of each UE was considered. We analyzed the UE individual satisfaction ratio per frame, by recording how many UEs met their QoS requirements in the simulation time. The details are described as follows:

- **System**: The QoS satisfaction ratio of the system. The ratio was defined as the percentage of UEs that met their QoS requirements.
- **VoIP:** The QoS satisfaction ratio of UEs generating voice traffic (GBR).
- **Video:** The QoS satisfaction ratio of UEs generating video traffic (MBR).
- **MUE:** The QoS satisfaction ratio of MUEs.
- **PUE:** The QoS satisfaction ratio of PUEs.
- **RE:** The QoS satisfaction ratio of UEs in the coverage of the CRE signal. A positive CRE bias 6 dB was added to the downlink received signal strength from pico BS.

### 4.4. Simulation Results

#### 4.4.1. Evaluation for the Cell Selection of HCI Scheme

The initial distribution of UE population in Macro BSs and pico BSs were 60% and 40%, respectively. To compare the HCI scheme with the SINR cell selection scheme, we let Macro BSs have higher load in order to assess whether pico BSs could offload the bandwidth requirements from UEs to Macro BSs. Figure 8 shows the population ratio of MUEs and PUEs of the two cell selection schemes. In the HCI scheme, more UEs could be offloaded to pico BSs from Macro BSs due to the consideration of the cell load. As a result of load balancing between Macro BSs and pico BSs, Figure 9 shows that the HCI scheme was able to achieve better QoS satisfactions from different aspects, in particular the QoS satisfaction of the whole system was increased by 3%. Notably, there was one exception, which was the PUE's QoS satisfaction. Clearly, since pico BSs were less loaded, the HCI scheme associated more UEs to pico BSs which resulted in less QoS satisfaction for PUEs as compared to the SINR scheme.

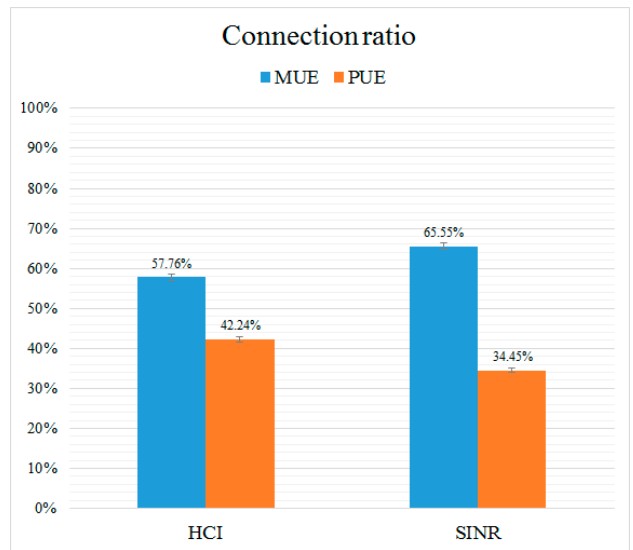

**Figure 8.** Connection ratio of the Macro user equipment (MUE) and Pico user equipment (PUE).

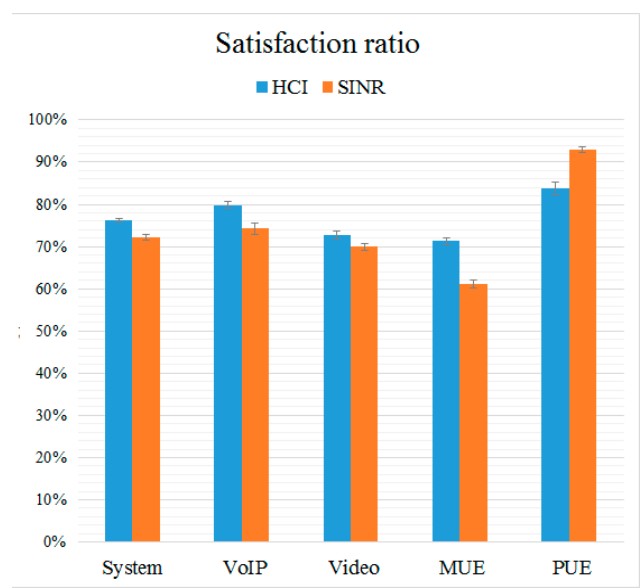

**Figure 9.** Satisfaction ratio of the different cell selection schemes.

#### 4.4.2. Evaluation for the Resource Allocation of MTS Scheme

In this scenario, the distribution of UE population was the same as that of the cell selection scenario. We verified our proposed MTS scheme by comparing to the original max-min scheduling. As shown in Table 3, the max-min scheme can achieve almost optimal radio resource scheduling. In the MTS scheme, we added a threshold to protect the MUEs' QoS in dual connectivity mode. Figure 10 shows that the MTS scheme yielded higher QoS satisfactions from all aspects. Especially, the MTS scheme yielded 5.07% higher QoS satisfaction of the whole system as compared to the max-min scheme.

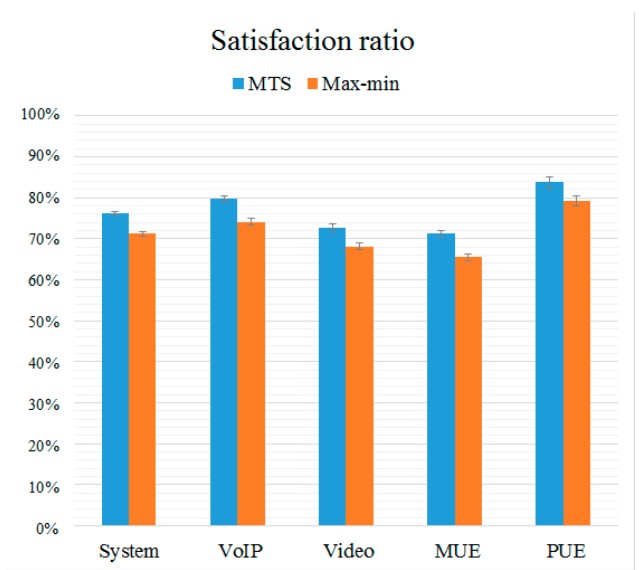

**Figure 10.** Satisfaction ratio of the different scheduling schemes in scenario 2.

#### 4.4.3. Evaluation for the Resource Allocation of PF Scheme

Scenario 3 was designed to show the effectiveness of the MTS scheme as compared to the PF scheduling. In addition, it also tries to demonstrate that the PF scheduling is not suitable in dual connectivity (DC) mode. Figure 11 compares the QoS satisfaction of the MTS scheme in DC mode as well as that of the PF scheme in both DC and signal connectivity (SC) modes. Clearly, the MTS scheme

outperformed the PF scheduling, in particular, it outperformed the PF scheme by at least 34.55% in the QoS satisfaction of the whole system. Since the PF scheduling is based on the fairness principle among UEs, it suffered the disadvantage that some UEs with poor CQI may have participated in competition of the radio resources. As a consequence, it was interesting to observe that the simulation results showed that the system satisfaction of the PF scheme was better in the SC mode than that in the DC mode.

While the MTS scheme achieved better QoS satisfaction, we may wonder its fairness. The fairness index of a scheme is defined as follows.

$$\text{Fairness index} = \frac{\left(\sum_{i=1}^{n} x_i\right)^2}{n \times \sum_{i=1}^{n} x_i^2} \tag{22}$$

where $x_i$ is the throughput of the $i$th UE and $n$ is the number of UEs in the system.

Table 7 shows the fairness index of the MTS in DC mode as well as the PF scheduling in both DC and SC modes. We observed that PF in SC mode had the best fairness index, while the MTS in DC mode had the lowest fairness index. However, the difference was not very significant. In other words, the MTS scheme could achieve much higher QoS satisfaction by sacrificing a little fairness as a tradeoff.

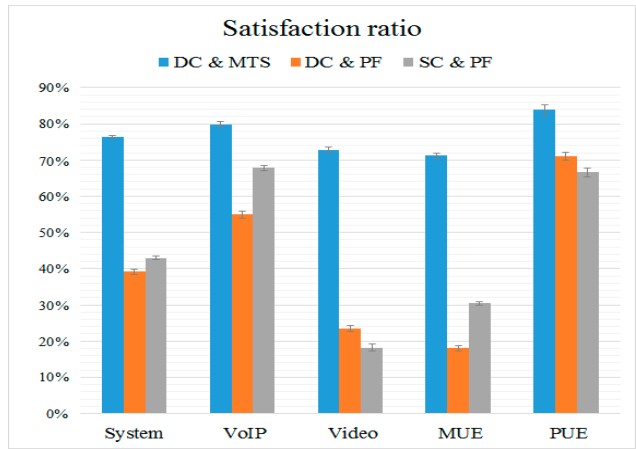

**Figure 11.** Satisfaction ratio of the different scheduling schemes in scenario 3.

**Table 7.** Fairness index of the Max–min Threshold Scheduler (MTS) scheme and the Proportional Fair Scheduler (PF) scheduling.

|  | MTS in DC Mode | PF in DC Mode | PF in SC Mode |
|---|---|---|---|
| Fairness index | 0.50623 | 0.64226 | 0.68495 |

4.4.4. Evaluation for the E-ICIC of QD-ABS Scheme

Scenario 4 compares the performance of the Q-learning-based dynamic ABS (QD-ABS) scheme with that of the static ABS scheme. For the static ABS scheme, the ABS ratio was set to 0.5 according to most of the literature. Recall that the QD-ABS scheme adopts the Q-learning mechanism to dynamically adjust the ABS ratio according to the feedback of the system. Two scenarios were simulated, we first simulated the case where more UEs were MUEs. Specifically, the UE population of Macro BS was three times more than that of pico BS. In the second case, all UEs were evenly distributed.

Figure 12 presents that the system QoS satisfaction ratio of the QD-ABS scheme was 4.06% higher than that of the Static ABS scheme in the first scenario. In the second scenario where UEs were evenly distributed, the proposed QD-ABS scheme was still better than the static ABS scheme, as shown in Figure 13.

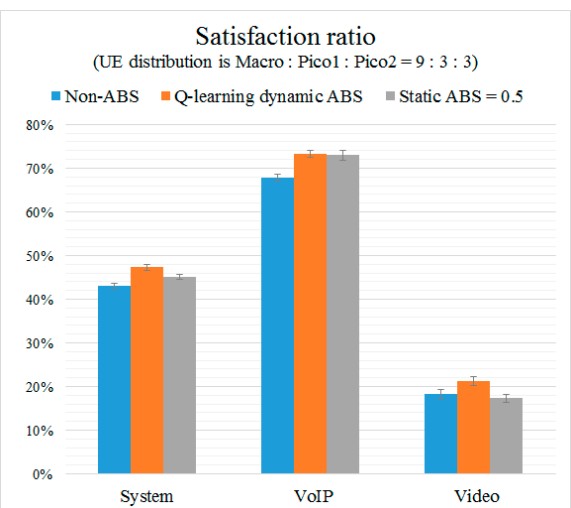

**Figure 12.** Satisfaction ratio of the different enhanced inter-cell interference coordination (e-ICIC) schemes in the overload situation of the Macro BS.

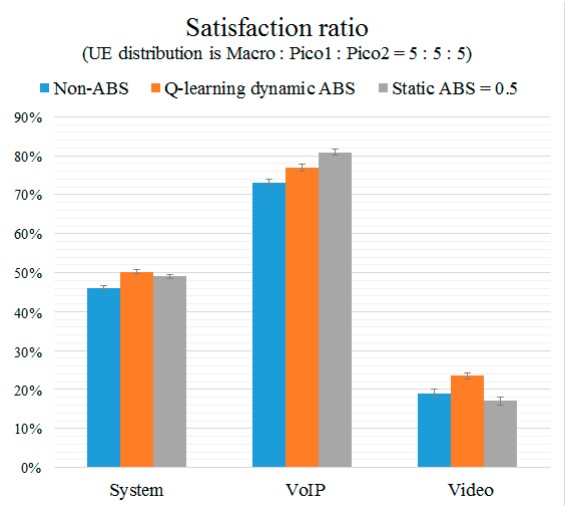

**Figure 13.** Satisfaction ratio of the different e-ICIC schemes when UEs are evenly distributed.

### 4.4.5. Evaluation for the Dual Connectivity Combined with CRE and ABS

Both the dual connectivity and the e-ICIC can improve the performance of the UE at the edge of BSs. For this reason, we propose a comprehensive scheme to explore the feasibility of combining the dual connectivity with CRE and ABS, and evaluate whether it can effectively improve the QoS bandwidth requirements of edge UEs via a scenario 5 simulation. Since the comprehensive scheme is based on the connection mode, cell selection, resource allocation, e-ICIC, and CRE bias, we compared its performance with individual schemes, including SC, DC, non-ABS and Static ABS, and Q-learning dynamic ABS.

In this scenario, we focused on the dual connectivity mode. Notably, the DC mode with CRE and ABS did not yield better system QoS satisfaction, but it was useful to increase the QoS satisfaction of edge UEs. Figure 14 shows the QoS satisfaction of different combinations of schemes of different aspects where the labels RE, RE VoIP, RE Video on the x-axis denote the UEs in the CRE region, the UEs of type GBR in the CRE region, and the UEs of the type MBR in the CRE region respectively. UEs in CRE region are referred to as cell edge UEs. We can observe from Figure 14 that by integrating the mechanisms of dual connectivity, CRE, and ABS, the QoS satisfaction ratio of cell edge UEs can be improved by 10.76% as compared to the traditional approach.

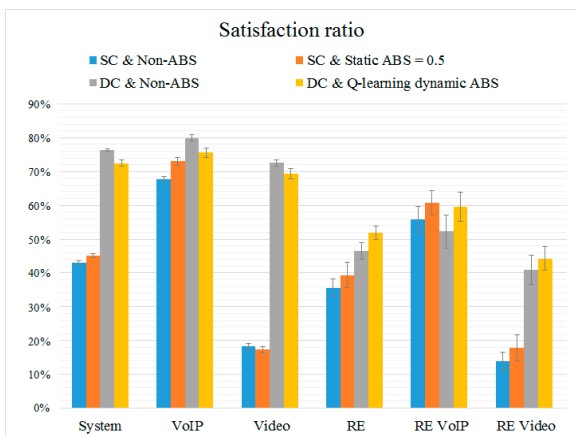

**Figure 14.** Satisfaction ratio of the scheme that integrates dual connectivity with cell range expansion (CRE) and ABS.

## 5. Conclusions and Future Works

In this work, we discussed the development of spectrum utilization in the future. For heterogeneous network research, we designed how to integrate dual connectivity, cell selection, resource allocation, and e-ICIC in the co-channel networks. On the other hand, we studied UEs with different QoS requirements and traffic characteristics, such as voice traffic (low data transfer rate) and video traffic (high data transfer rate). In the resource allocation mechanism, we propose the MTS scheduling to meet the QoS requirement, considering the resource integration across the base stations to enhance the data transfer rate. In the cell selection mechanism, we propose the HCI scheme to consider not only the signal quality received by the UE, but also the remaining RBs of each base station, and further achieve the load balancing between the base stations. In the e-ICIC mechanism, we propose the QD-ABS scheme to adjust the ABS ratio dynamically according to system feedback.

The ultimate goal of the proposed mechanisms is to improve the QoS satisfaction ratio of UEs. Our simulation results showed that our MTS scheduler was able to achieve 31.44% higher rate than the Proportional Fairness scheduler; our HCI cell selection scheme yielded 2.98% higher rate than the SINR cell selection scheme; the QoS satisfaction ratio of our Q-learning-based dynamic ABS scheme was 4.06% higher than that of the static ABS scheme. Finally, by integrating the mechanisms of dual connectivity, CRE, and ABS, the QoS satisfaction ratio of cell edge UEs could be improved by 10.76% as compared to the traditional approach.

We are expecting to see some small-scale commercial deployments for the fifth generation (5G) networks by some leading operators in 2019 worldwide. The 5G air interface, called new radio (NR), is expected to interwork with different wireless technologies where UEs will have multiconnectivity capabilities. In particular, in the early stage of 5G deployment, interworking with LTE-A based on dual connectivity will be necessary to boost the deployment and has been proposed by 3GPP as a 5G operational requirement [2,36]. Extending the dual connectivity to multi-connectivity in 5G will become an important technique and require further study [37].

**Author Contributions:** Conceptualization, R.-H.H. and M.-C.P.; methodology, R.-H.H. and K.-C.C.; software, K.-C.C.; validation, R.-H.H. and M.-C.P.; formal analysis, R.-H.H. and K.-C.C.; investigation, R.-H.H. and M.-C.P.; resources, R.-H.H.; data curation, K.-C.C.; writing—original draft preparation, K.-C.C.; writing—review and editing, R.-H.H. and M.-C.P.; visualization, K.-C.C.; supervision, R.-H.H.; project administration, R.-H.H.; funding acquisition, R.-H.H.

**Funding:** This research was funded by Ministry of Science and Technology, Taiwan, grant number MOST 105-2221-E-194-03 and MOST 106-2221-E-194-021-MY3.

**Conflicts of Interest:** The authors declare no conflict of interest.

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
