# Peer review of "QoS-Guaranteed Radio Resource Management in LTE-A Co-Channel Networks with Dual Connectivity"

_applsci, doi:10.3390/app9153018_

Round 1
Reviewer 1 Report
A major revision is needed. In particular, several important aspects are unclear or not described in the paper:
1) What are the traffic patterns for VoIP and Video? How much is the load provided by these applications compared to Macro BS and Pico BS capacity?
2) How does the SINR map look like for the proposed BS positions scenario? Do Pico BSs interfere with Pico BSs in the neighboring hexagonal sector?
3) Why only a value of 0.5 for ABS density is chosen for the reference ABS solution? Is it the best one among all possible static configurations (for the considered in the paper scenario)? Also, how fast does the Q-learning approach converge to some value (or it does not converge to a nearly constant value)?
4) How to calculate the number of required RBs according to equation (5)? Specifically, it is not clear how to calculate the data rate of RB since different RBs have different data rates (because of interference/fading)? Is it the minimal/average data rate?
5) The Max-min approach proposed is paper is very similar to the simple "best CQI" approach with the difference that the proposed approach also considers QoS requirements for users. However, the comparison is provided only for PF, which is not aimed at throughput maximization. Figures for Best CQI will emphasize the advantages of the proposed approach (if there will be some gain).
While the paper contains informative satisfaction ratio bar plots showing the gain provided by the proposed solution, it will be interesting to see latency, throughput plots for different solutions, and also the load (e.g., the percentage of allocated resource blocks) for Macro BS and Pico BSs.
In addition, there are several papers about dynamic ABS density, both with learning approach (see, for example, A Fuzzy Q-Learning Approach for Enhanced Intercell Interference Coordination in LTE-Advanced Heterogeneous Networks) and without it (Dynamic Protected-Subframe Density Configuration in LTE Heterogeneous Networks). The paper will benefit from the comparison of the proposed dynamic ABS adjustment approach with other approaches from the literature.
Reviewer 2 Report
In this paper, the authors propose dynamic QoS provisioning technique for LTE networks using Q-learning, which is a kind of machine learning. However, in order to implement this proposal, it is recommended to specify some of the following issues.
1. In section 3.3.1 and 3.3.2, the reinforcement learning and q-learning are examples of existing research, so I recommend that the authors add a 'related works' chapter at the beginning and explain it separately.
2. The resource requirements of each terminal can change from time to time. How often, in what way, does the proposed technique reflect this? (The description shall be added in Section 3.)
3. It is necessary to precisely define and calculate the proposed parameters and explain the update cycle of them. In the paper, they are defined very abstractly.
- Line 402, gain factor
- Line 404, ABS
- Line 413, satisfaction ratio
- Line 415, affected throughput
4. In Line 442, the authors shall provide a clear definition/condition of "suitable ABS"
5. In line 549, the authors shall describe the definition and calculation method of the fairness index
Reviewer 3 Report
This paper is investigating resource allocation and management problem in case dual connectivity is activated for user equipments of LTE-A. The authors compare various scheduling approaches together with their own solutions of max min threshold scheduler as well as utilize Q- learning approach to adjust the paramters of ABS ratio to maximize the overall system utility. The authors have also validated their contributions via MATLAB simulations.
The paper’s contributions are suitable for publications as a journal. The paper is written well with good organization structure. Some minor comments:
- On page 3 line 125, sentence starting with “By integrating..”is not complete.
- On page 3 line 102 “QoS requirements are considered”
- On page 2 line 56, “including resource allocation,”
- On page 5 line 183, “which will be described…”
- On page 6 line 200, “As can be observed, Random method…”
- On page 7 line 252, “used to assign…”
- On page 10 line 354, “Macro BSs to synchronize…”
- On page 11 line 392, “The first one is that all of the MUE must satisfy…”
- On page 17 line 548, it should be “Table 6”
- On page 18 line 579, “..including SC, DC,”
Reviewer 4 Report
Due to the mathematical expressions involved, please refer to the review comments in the pdf file attached.

Author Response
Thank you very much for the very valuable comments, please see the attached file for our responses.

Round 2
Reviewer 2 Report
The authors sincerely improved the quality of the paper by reflecting the reviewer's opinions. I think this paper can be accepted.Reviewer 4 Report
The issues mentioned previously are addressed.